# Oxyresveratrol Attenuates Inflammation in Human Keratinocyte via Regulating NF-kB Signaling and Ameliorates Eczematous Lesion in DNCB-Induced Dermatitis Mice

**DOI:** 10.3390/pharmaceutics15061709

**Published:** 2023-06-12

**Authors:** Hung Gia Tran, Aussavashai Shuayprom, Patipark Kueanjinda, Asada Leelahavanichkul, Prapai Wongsinkongman, Siriwan Chaisomboonpan, Apiwat Tawatsin, Kriangsak Ruchusatsawat, Jongkonnee Wongpiyabovorn

**Affiliations:** 1Graduate Program in Clinical Sciences, Faculty of Medicine, Chulalongkorn University, Bangkok 10330, Thailand; tghung@ctump.edu.vn; 2Department of Medical Sciences, Ministry of Public Health, Nonthaburi 11000, Thailand; aussavashai.s@dmsc.mail.go.th (A.S.); prapai.w@dmsc.mail.go.th (P.W.); siriwan.c@dmsc.mail.go.th (S.C.); apiwat.t@dmsc.mail.go.th (A.T.); ruchusatsawat@gmail.com (K.R.); 3Center of Excellence in Immune-Mediated Diseases, Department of Microbiology, Faculty of Medicine, Chulalongkorn University, Bangkok 10330, Thailand; patipark.k@chula.ac.th; 4Center of Excellence on Translational Research in Inflammation and Immunology (CETRII), Faculty of Medicine, Chulalongkorn University, Bangkok 10330, Thailand; aleelahavanit@gmail.com

**Keywords:** oxyresveratrol, skin inflammation, eczema, dermatitis, anti-inflammation, anti-proliferation

## Abstract

Oxyresveratrol (ORV) is one of the novel antioxidants having been extensively studied in recent years. One of the main sources of ORV is *Artocarpus lakoocha*, which has been used in traditional medicine in Thailand for decades. However, the role of ORV in skin inflammation has not been clearly demonstrated. Therefore, we investigated the anti-inflammatory effects of ORV on dermatitis model. The effect of ORV was examined on human immortalized and primary skin cells exposed to bacterial components including peptidoglycan (PGN) and lipopolysaccharide (LPS) and 2,4-Dinitrochlorobenzene (DNCB)-induced dermatitis mouse model. PGN and LPS were used to induce inflammation on immortalized keratinocytes (HaCaT) and human epidermal keratinocytes (HEKa). We then performed MTT assay, Annexin V and PI assay, cell cycle analysis, real-time PCR, ELISA and Western blot in these in vitro models. H&E staining, immunohistochemistry (IHC) staining with CD3, CD4 and CD8 markers were used to evaluate the effects of ORV in in vivo model of skin inflammation using BALB/c mice. Pretreatment of HaCaT and HEKa cells with ORV inhibited pro-inflammatory cytokine production through inhibition of NF-κB pathway. In DNCB-induced dermatitis mouse model, ORV treatment reduced lesion severity, and skin thickness and numbers of CD3, CD4 and CD8 T cells in the sensitized skin of mice. In conclusion, it has been demonstrated that ORV treatment can ameliorate inflammation in the in vitro models of skin inflammation and in vivo models of dermatitis, suggesting a therapeutic potential of ORV for treatment of skin diseases particularly eczema.

## 1. Introduction

Oxyresveratrol (2,4,3′,5′-tetrahydroxystilbene; ORV) is a phytoalexin present in large amounts in the heartwood of *Artocarpus lakoocha* [1], and the root of *Morus alba* [2]. ORV and its derivatives have garnered considerable research interest over the years due to their antioxidant, antibacterial, antiviral, and anticancer properties [3]. In addition, ORV has no effect on cell viability of macrophages and glial cells at concentrations higher than resveratrol (RSV), the previous well-known antioxidant [4,5]. Studies on various cell types have revealed that, in addition to its antioxidant properties, ORV has anti-inflammatory and anti-proliferative properties. In lipopolysaccharide (LPS)-induced inflammation model of macrophage, ORV can inhibit inducible nitric oxide synthase (iNOS) expression in dose-dependent and NF-κB-dependent manners [4,6]. Using at moderate concentration (40 μM/L), ORV can effectively inhibit IL-6 production in RAW 264.7 cell, a macrophage cell line [6]. Additionally, ORV significantly reduces the release of IL-6 and MCP-1 from IL-1β-stimulated in human microglial cells (HMC3), an in vitro neuroinflammation model [7]. Along with the protecting effect on microglial cells, ORV also preserve the spinal cord’s integration in a model of spinal cord injury murine [5]. In addition to macrophage-associated diseases, ORV exhibits effects on many cancer cell lines; antiproliferative effect in hepatic cancer cell line HepG2, breast cancer cell line MCF-7, and colon cancer cell line HT-19 and inhibition of cell survival in squamous cell carcinoma cell lines (i.e., HSC-3, HN-8, and HN-30) [8,9,10].

Skin is considerably the largest organ of the body that functions as a barrier protecting the body from the environment that consists of harmful chemical, physical, and pathogenic agents. Cutaneous diseases have the highest prevalence compared to other diseases worldwide [11,12]. The proportion of cutaneous disorders may vary by countries, ethnic and ages. To assess the influence of skin diseases on life, a systematic study using the disability adjusted life year (DALY) metric showed that the burden of inflammatory skin condition had the first position among the other types of skin conditions. Particularly, inflammatory skin condition accounted for about 0.38 year of life lost [13]. Ranking among the health problems worldwide with all ages, eczema was considered as the prominence, with 11–13% for almost ethnicities [14]. Among inflammatory skin diseases, eczematous skin condition is one of the most prevalence. Eczematous skin conditions share inflammatory characteristics, for example, production of proinflammatory cytokines such as IL-1, IL-6, IL-8, and TNF-α that further stimulate the common inflammatory signaling pathways, such as NF-κB and JAK/STAT pathways [15,16,17].

Managements of eczematous skin lesions aim to diminish inflammation and restore skin barrier. Corticosteroids are often used as a cornerstone drug because of their effectiveness in anti-inflammatory, antiproliferative, and immunosuppressive properties. Typically, treatment of skin lesions requires both oral and topical administration [18]. An early administration of corticosteroids can facilitate the restoration of filaggrin, loricrin, and trans-epidermal water loss (TEWL) to optimal levels to restore skin barrier. Treatment with corticosteroid can also effectively aid in epidermal hydration [11]. However, there are a number of side effects reported from long-term use of corticosteroids. Alternatively, calcineurin inhibitors, such as tacrolimus and pimecrolimus which can inhibit T cell activation, have been used to treat moderate and mild dermatitis, particularly atopic dermatitis [19]. In addition to controlling skin inflammation, ongoing treatment can alter skin microbiome, as demonstrated by our previous study which found that monotherapy with tacrolimus could reverse dysbiosis in atopic dermatitis patients, with the commensal genera, such as *Dermacoccus* sp., *Pseudomonas* sp., *Corynebacterium* sp., *Proteus* sp., *Micrococcus luteus*, and *Lactobacillus aureus*, became more prevalent [20]. Other therapeutic options for eczematous skin lesions also include moisturizers, antibiotics, immunosuppressive agents, and biologics, especially for atopic dermatitis whose pathophysiology is complex and multifactorial [21,22,23,24,25,26].

Because reducing inflammation has been one of the main objectives for long-term control of eczematous skin conditions, anti-inflammatory agents should be able to inhibit key inflammation regulators and have minimal adverse effect on eczema patients. In this study, we therefore examine therapeutic properties of ORV and its efficacy in vitro using primary human keratinocytes and in vivo using mouse model of eczematous skin conditions.

## 2. Materials and Methods

### 2.1. Culture of HaCaT Cells and Primary Keratinocyte HEKa Cells

HaCaT was maintained in Dulbecco’s Modified Eagle Medium (DMEM, high glucose, pyruvate; 11995), supplemented with 10% fetal bovine serum (#10270), 0.01 M HEPES (#15630) and 100 U/mL penicillin-streptomycin (#15140) and was incubated at 37 °C in humidified incubator containing 5% CO_2_. All the media and reagents for HaCaT cell culture were purchased from Gibco (Grand Island, NY, USA). A night before treatment process, the cells were maintained in serum-free and antibiotic-free media. Cells were then pre-treated with ORV for 24 h before exposed to PGN (Sigma-Aldrich, St. Louis, MO, USA) for 24 h.

Human Epidermal Keratinocytes (HEKa) derived from normal human adult foreskin were purchased from ATCC (PCS-200-011). HEKa cells were cultivated in dermal cell basal medium containing supplements from a keratinocyte growth kit (ATCC, PCS-200-040) and incubated at 37 °C in a humidified incubator containing 5% CO_2_. Cells were then treated as described and collected for further experiments.

### 2.2. Reagents

#### 2.2.1. Oxyresveratrol (ORV)

ORV (2,4,3′,5′-tetrahydroxystilbene) extracted from *Artocapus lakoocha*, was kindly given by the Thailand’s Ministry of Public Health. The purity of ORV was analyzed by using ultra-performance liquid chromatography and had a purity greater than 95%. Commercial ORV purchased from Sigma-Aldrich (St. Louis, MO, USA) was used for comparison. For in vitro cell culture experiment, the compound was dissolved in DMSO and diluted with PBS to achieve the desired working concentration, whereas the final concentration of DMSO in the solution was 1%. For in vivo experiment in dermatitis mouse model, a cream containing ORV, which was also developed by the Ministry of Public Health and Chulalongkorn University’s Faculty of Medicine, were determined for an optimal formulation using DNCB-induced dermatitis mouse model.

#### 2.2.2. Peptidoglycan (PGN) and Lipopolysaccharides (LPS)

PGN of *Staphylococcus aureus* (#77140) and LPS of *Escherichia coli* O111:B4 (#L4391) were purchased from Sigma-Aldrich (St. Louis, MO, USA). Each compound was dissolved in sterile distilled water and further diluted with PBS for in vitro cell culture experiment.

### 2.3. Cell Viability Assay

To detect the cytotoxicity effect of ORV on HaCaT and HEKa cell viability, we seeded 0.5 × 10^4^ cells/well in a flat bottom 96-well plate (SPL Life Sciences, Gyeonggi-do, Korea) with DMEM containing 10% FBS, 1% penicillin and streptomycin at 37% in humid 5% CO_2_ atmosphere for 24 h. Then, cells were treated with ORV at several concentration for 24 h. 50 μL MTT reagent (3-(4,5-dimethylthiazol-2-yl)-2,5-diphenyltetrazolium bromide) (ab211091, Abcam Plc., Aibo Trading Co., Ltd., Shanghai, China) was added to each well and incubate in 5% CO_2_ incubator at 37 °C for 3 h. Next, optical densities (OD) were measured at a wavelength of 490 nm using Varioskan microplate reader (Thermofisher Scientific, Grand Island, NY, USA).

### 2.4. Apoptosis Assay

The apoptosis assay was performed on HaCaT cells using APC Annexin V Detection kit with PI (#640932, BioLegend, San Diego, CA, USA) according to the manufacturer’s instructions. Before analysis, HaCaT cells were seeded at 0.5 × 10^6^ cells/well in a 6-well plate (SPL Life Sciences, Gyeonggi-do, Republic of Korea) and incubated overnight. On the next day, the cells were treated with ORV for 24 h. The cells were then trypsinized and washed twice with Cell Staining buffer (#420201, Biolegend, San Diego, CA, USA). Annexin V binding buffer at a concentration of 1 × 10^6^ cells/mL was added to the cells that were later stained with APC annexin V and propidium iodine (PI) for 15 min at room temperature. Cells were measured for annexin V and PI intensity suing BD^®^ LSR II flow cytometer (BD Biosciences, Franklin Lakes, NJ, USA) and the data were analyzed using FlowJo 10 cytometry analysis software (FlowJo, Ashland, OR, USA).

### 2.5. Cell Cycle Analysis

For cell cycle analysis, HaCaT cells were trypsinized after treatment with ORV for 24 h. The cells were then washed with PBS buffer and fixed using 70% ethanol at −20 °C. After cell fixation, the cells were washed twice with cold PBS buffer and treated with 50 μL of 100 μg/mL RNAse to remove RNA. Finally, the cells were stained with 200 μL of Propidium Iodine (PI) and collected data of PI-stained cells using BD LSR II flow cytometer (BD Biosciences). Data were analyzed by FlowJo 10 cytometry analysis software (FlowJo, Ashland, OR, USA).

### 2.6. RNA Isolation, Complementary DNA Synthesis, PCR and Quantitative Real Time-PCR Analysis

Total RNA from cells and mice skin specimens was isolated using TRIzol reagent (Gibco, Carlsbad, CA, USA) according to the manufacturer’s instructions. Subsequently, the first strand of cDNA was synthesized from a starting concentration of total RNA at 500 ng using iScript^TM^ cDNA Synthesis kit (#1708891, Biorad, Hercules, CA, USA). PCR was performed using EmeraldAmp^®^ GT PCR Mastermix (RR310A, Takarabio, Kusatsu, Shiga, Japan). The optimal PCR condition was as follows: 98 °C for 10 s, 60 °C for 10 s and 72 °C for 60 s for 30 cycles. For Real time-PCR, the amplification steps were performed using PowerUp^TM^ SYBR^TM^ Master Mix (A25742, Applied Biosystem, Vilnius, Lithuania) as follows: 1 cycle at 95 °C for 10 min, 40 cycles at 95 °C for 30 s, 60 °C for 30 s, and 72 °C for 1 min (see Appendix A).

### 2.7. IL-6 and IL-8 Cytokine Measurement

The levels of supernatant cytokines were measured using human IL-6 and IL-8 ELISA kits (#88-7066 and #8086-22, respectively; Thermofisher Scientific, Carlsbad, CA, USA). Cell cultures were performed in triplicates under each treatment condition and their supernatants were collected and processed for cytokine detection according to the manufacturer’s instructions. The optical densities were measured at 450 nm and 570 nm with Varioskan Microplate reader (Thermofisher Scientific, Grand Island, NY, USA). The concentrations were calculated from the standard curve generated by a curve-fitting program.

### 2.8. Western Blot Analysis

Western blot analysis was utilized to detect intracellular proteins, NF-κB, phosphorylated-NF-κB, and cleaved caspase-3. Cells were lysed for protein lysate using RIPA buffer (#9806, Cell Signaling Technology, Danvers, MA, USA) containing protease/phosphatase inhibitor cocktail (#5872, Cell Signaling Technology, Danvers, MA, USA). For detection of GAPDH, NF-κB, and phosphorylated-NF-κB, 20 μg of protein lysate was used. For detection of cleaved caspase-3, 50 μg of the protein lysate was used. The protein lysates were loaded into 12% SDS-PAGE and run at 100 volts for 90 min to separate the proteins. After that, the proteins were transferred to nitrocellulose membranes (Amersham, Arlington Heights, IL, USA) at 70 volts for 75 min. The membranes were incubated in blocking solution for 60 min and then in blocking solution containing the following antibodies: NF-κB p65 (D14E12) XP rabbit mAb (#8242), Phospho-NF-κB p65 (Ser536) (93H1) rabbit mAb (#3033), Cleaved caspase-3 (Asp175) (5A1E) rabbit mAb (#9664), and GAPDH (D16H11) XP rabbit mAb (#5174). Then, the membranes were washed twice in PBS buffer to remove excessive antibodies and incubated in blocking buffer containing HRP-conjugate anti-rabbit secondary antibody (#7074) followed by washing steps. All antibodies were purchased from Cell Signaling Technology (Danvers, MA, USA). The protein bands were detected using UltraScience Pico Plus Western Substrate (Bio-Helix, New Taipei City, Taiwan). GAPDH expression levels were used as internal control. The band intensity of proteins of interest was quantified from triplicate Western blot images using Image Lab software version 6.1 (Bio-Rad Laboratories, Hercules, CA, USA).

### 2.9. Dermatitis Animal Model

Seven-to-nine-week-old BALB/c mice were purchased from Nomura Siam (Bangkok, Thailand). The animals were maintained according to the standard animal care protocol approved by the Animal Care and Use Committee of the Faculty of Medicine, Chulalongkorn University (No. 032/2565; approved on December 2022). Mice were housed in a husbandry unit with 12 h light/dark cycle under Thermos-regulated (22 ± 2 °C) and humidity-controlled (50 ± 10%) condition and provided with standard diet and water ad libitum. All mice were randomly assigned into four groups: no treatment, DNCB, DNCB + cream base, DNCB + ORV, consisting of 3 mice in each group.

Dermatitis was induced in mice by application of 2,4-Dinitrochlorobenzene (DNCB) on their skin as described previously with some modifications [27,28]. Briefly, the dorsal hairs of mice were completely removed. And on the next day, 100 μL of 1% DNCB in acetone/1 cm^2^ were applied twice a week at the bare skin to sensitize the mouse skin. Then, 100 µL of 0.5% DNCB was applied to the sensitized skin areas twice a week to induce dermatitis. In ORV treatment group, ORV was applied regularly on the skin. On day 23, mice were sacrifice and skin specimens of the mice were collected for pathological and immunohistochemistry assessments.

### 2.10. Evaluation of Severity Using Clinical Skin Score

Manifestations of dermatitis were collected twice per week for four weeks. Four symptoms, including erythema/hemorrhage, scarring/dryness, edema, and excoriation/erosion, were graded on a scale from 0 to 3 (none, 0; mild, 1; moderate, 2; severe, 3) to determine the severity of dermatitis [27]. The clinical skin score, ranging from 0 to 12, was defined as the sum of the scores from each symptom.

### 2.11. Pathology Observation and Immunohistochemistry

The dorsal skins were fixed in 4% paraformaldehyde for 24 h before paraffinization. The paraffin-embedded skin sections were then embedded on glass slides, and they were stained with hematoxylin and eosin (H&E) to examine skin histology and skin thickness. Immunohistochemistry staining was performed with CD3, CD4, and CD8 markers to identify the recruitment of lymphocytes into the skins.

### 2.12. Statistical Analysis

Statistical analysis was performed using GraphPad Prism version 9.0. For multiple comparison test, one-way ANOVA was utilized. A pairwise comparison was also used to test the statistical significance of different ORV concentration groups of samples. Unless otherwise stated, the data included measurements collected from triplicate samples, and their numerical values were presented in graphical formats as mean ± standard deviation (SD). The difference is considered statistically significant when the *p* value is less than 0.05.

## 3. Results

### 3.1. Effects of Oxyresveratrol on Human Immortalized (HaCaT) and Primary (HEKa) Keratinocytes

To investigate the effect of ORV on skin, in vitro experiments on immortalized and primary keratinocytes (HaCaT and HEKa, respectively) were conducted. After treating the cells with various concentrations of ORV, cell viability was determined. There was no significant difference in the cell viability of HaCaT between our ORV extract and the commercial ORV. The IC_50_ values of ORV for the two cell types HaCaT and HEKa were 81.68 μg/mL and 266.8 μg/mL, respectively (Appendix A). Until the ORV concentration reached 10 μg/mL, neither type of keratinocyte died. Therefore, all of the following experiments were conducted with ORV concentrations less than 10 μg/mL unless stated otherwise.

After determining the optimal concentration for in vitro experiments, we further investigated the proliferation and apoptosis effects of ORV on HaCaT. HaCaT cells were treated with ORV at concentrations of 1, 5, and 10 μg/mL for 24 h before being stained for apoptosis markers with annexin V and propidium iodine (PI). Apoptotic cells in both early and late apoptosis increased approximately two-fold in all ORV-treated groups compared to the control group, according to flow cytometer analysis of annexin V and PI stained cells (Figure 1a; quadrant (Q) 2 indicates late apoptosis and Q3 indicates early apoptosis). However, there was no statistically significant difference in the number of apoptotic cells between cell groups treated with different ORV concentrations.

Because ORV inhibited cell growth, we wanted to know which cell cycle phase it inhibited. To answer this question, we used PI staining of HaCaT cells after treatment with various concentrations of ORV and performed cell cycle analysis. After treatment with 1 μg/mL, 5 μg/mL, and 10 μg/mL of ORV, cells were arrested at G0/G1 (76.83 ± 0.83%), G2/M (14.20 ± 0.36%), and S phase (11.33 ± 0.25%), respectively, indicating that there was a shift in cell cycle arrest in HaCaT cells as the concentration of ORV increased (Figure 1b). To determine how ORV affected cell death, we looked at the cleaved form of caspase-3, a key initiator of apoptosis signal, in HaCaT cells after 24 h of treatment with 5 μg/mL of ORV. The level of cleaved caspase-3 was nearly threefold higher (3.29 ± 0.18, *p* value < 0.05) in the ORV treatment group than in the untreated group (Figure 1c). According to our findings, ORV may activate caspase-3 and induce apoptosis, as evidenced by cell cycle arrest in human keratinocytes, which can shift depending on ORV concentration.

### 3.2. Anti-Inflammatory Effect of Oxyresveratrol on HaCaT Cells

Microbial colonization on the skin is one of the causes of skin inflammation. Once the normal flora population is disrupted, the inflammatory microbial population can overwhelm. Bacterial cell wall components such as PGN and LPS can bind to their cognate receptors on the cells composed of the skin, thereby stimulating keratinocyte inflammation. After assessing effect of ORV on cell proliferation, we next investigated its effect on inflammatory regulation in both keratinocyte cell line (HaCaT) and primary keratinocyte (HEKa). HaCaT cells were stimulated with varying concentrations of PGN and monitored for the levels of pro-inflammatory cytokines. We found that 5 μg/mL of PGN was sufficient to induce expression of inflammatory cytokines in HaCaT cells without affecting the cell viability (Appendix A). To examine the effect of ORV on prevention of inflammation, HaCaT cells were pretreated with various concentrations of ORV before being stimulated with PGN. The results showed that pretreatment at 1, 5 and 10 μg/mL of ORV significantly inhibited IL-6 and IL-8 cytokine production at mRNA and protein levels (Figure 2a–d; *p* < 0.0001); however, there was no significant difference on the cytokine expression levels between the ORV-pretreated groups. In addition, there was also no significant difference in the suppressive effect of IL-6 and IL-8 mRNA expression between commercial ORV and our ORV extract (Appendix A).

Because NF-κB is one of the central regulators of inflammatory signaling cascades, we therefore sought to verify that ORV treatment inhibited NF-κB transcription factor activation in keratinocytes. Adding PGN to HaCaT cells significantly increased NF-κB transcription factor activation, as indicated by an increase in phosphorylated NF-κB p65 subunit (1.53 ± 0.10; *p* < 0.05) (Figure 2e). However, an ORV treatment prior to PGN stimulation significantly reduced the level of phosphorylated NF-κB in HaCaT cells (0.86 ± 0.31; *p* < 0.05) (Figure 2e).

### 3.3. Anti-Inflammatory Effect of Oxyresveratrol on Primary Epidermal Keratinocytes

Furthermore, the anti-inflammatory properties of ORV in primary human epidermal keratinocytes (HEKa) were examined. We found that PGN and LPS stimulated different profiles of pro-inflammatory cytokines in HEKa cells (Appendix A). LPS stimulation, for instance, increased the expression of *IL-1β*, *IL-8*, *hBD3*, and *LL37* genes and pretreatment with ORV significantly reduced their expression (Figure 3). Similar results were observed when the cells were pretreated with ORV before PGN stimulation (Figure 4). However, *IL-8* expression level was not suppressed by ORV pretreatment when the cells were stimulated with a high dose of PGN (10 μg/mL) (Figure 4b). In summary, our results showed that pretreatment with low doses of ORV prior to stimulation can significantly reduce the expression of pro-inflammatory cytokine genes.

### 3.4. Oxyresveratrol Treatment Alleviates Skin Inflammatory Manifestation on Mice

We previously showed that ORV could not only induce apoptosis but also inhibit inflammatory cytokine production in immortalized and primary human keratinocytes in vitro. Because of its clearly strong anti-inflammatory effect, we wanted to use ORV as a topical treatment for dermatitis. We used DNCB to induce skin inflammation in BALB/c mice, followed by regular applications of ORV-based emollient cream to the mice’s inflamed skin (Figure 5a, top panel). Clinical manifestations were documented and scored (see Materials and Methods). On day 8 to 11, we observed clinical manifestations of DNCB-induced dermatitis on the mouse skin, including erythema, scarring/dryness, edema, and excoriation/erosion. For the next 10 days, emollient cream with or without ORV was applied on the mouse skin once every 2 days. Finally, mice were sacrificed on day 23 and their skins were collected for pathological and immunohistochemical analysis. As expected, clinical scores were significantly lowered as demonstrated by alleviated clinical manifestations in mice regularly treated with ORV for 10 days (0.5 ± 0.7) compared to the group receiving base emollient cream without ORV (3.5 ± 0.7; *p* < 0.05) and to the non-treatment group (5.5 ± 0.7; *p* < 0.01) (Figure 5a, bottom panel).

In addition to severe clinical manifestations on the skin, epidermal thickness also increases as the skin becomes chronically inflamed. Pathological examination of the skin tissues from the DNCB-treated mice revealed significantly increased epidermal thickness when compared to that of the control group (Figure 5b). Topical treatment with base emollient cream alone, on the other hand, significantly reduced epidermal thickness of the skin. However, the epidermal thickness of mice topically treated with ORV-containing emollient cream was reduced to levels comparable to the control group.

The skin of dermatitis patients has been studied and reportedly found to be colonized with skin-infiltrating lymphocytes. T cell populations such as CD3^+^, CD4^+^, and CD8^+^ T cells can secrete cytokines that promote the inflammatory response and contribute to the chronic state of the disease. We performed immunohistochemical staining and identified CD3, CD4 and CD8 T cells in dissected mouse skin tissues. Mice treated with DNCB alone had a higher number of T cells in their skin compared to the control group (Figure 6). The number of T cells in the skin of mice treated with base emollient cream alone decreased significantly when compared to the DNCB-treated group. The number of T cells was even lower in the skin from mice treated with ORV-containing emollients, but not as low as the number of T cells found in the control group. These findings suggest that ORV treatment can help alleviate dermatitis symptoms by reducing clinical manifestations, skin thickness, and the number of skin-infiltrating lymphocytes.

## 4. Discussion

Eczema is a type of skin inflammation in which skin permeability and barrier functions are compromised. Inflammation is accompanied by a concurrent increase in epidermal proliferation and disruption of epidermal differentiation, resulting in clinical manifestations such as erythema, scales and lichenification [29]. Eczema is one of the most common dermatological disorders that requires both immediate treatment and a long-term management strategy. ORV has long been reported as an antioxidant reagent with potential dermatological applications [2,30]. In addition, it has anti-proliferative effects on cancer cell lines as well as anti-inflammatory effects on macrophages and human microglial cells [4,6,8,9,10]. In this study, the anti-inflammatory efficacy of ORV on dermatitis models in vitro and in vivo was investigated using human keratinocyte cells and BALB/c mice, respectively.

Our study found that ORV reduced cell viability proportionally to its concentration in both HaCaT and HEKa cells. It had no effect on the viability of either cell type at concentrations of 10 μg/mL and lower, which is consistent with previous studies in macrophage cell lines and HaCaT cells (40 μM = 9.76 μg/mL) [6,31]. Despite having low cytotoxicity, our ORV still induced cleaved caspase-3 in HaCaT cells. Interestingly, we observed a shift in cell cycle arrest in HaCaT cells after treatment with increasing doses of ORV. The cell cycle shift could be caused by ORV interference in cell proliferation at different levels, from DNA replication to cell division, which suggest that ORV is a cell cycle non-specific anti-proliferation compound [32]. Because our ORV showed apoptosis-induced anti-proliferation property, this property is in line with that of glucocorticoid, one of the most commonly used therapeutic reagents for dermatitis, suggesting the potential use of ORV to reduce cellular proliferation in dermatitis [33].

In addition to anti-proliferative properties, we also investigated anti-inflammatory properties of ORV because skin inflammation is commonly observed in dermatitis. Initially, the anti-inflammatory effect of ORV was investigated using an immortalized keratinocyte cell line (HaCaT). The immortalized cell line, resulting from in vitro modification, may have some different properties from primary keratinocyte cell [34,35]. Besides, it has been accepted that HEKa is a better model for studying human keratinocyte because it is the primary cell. Thus, the anti-inflammatory effect of ORV was further extensively studies using HEKa. Our results revealed that ORV inhibited a variety of pro-inflammatory cytokines in keratinocyte in vitro model. In particular, IL-6 and IL-8 cytokines, which are reportedly produced by keratinocytes during the acute phase of skin inflammation in dermatitis and other skin diseases, were significantly reduced [36]. The master regulator of inflammatory signaling, NF-κB, was also disrupted, as evidenced by a decrease in phosphorylation of NF-κB p65 subunit in ORV-treated keratinocytes. Although our results suggest that ORV may inhibit pro-inflammatory cytokines via canonical NF-κB p65 signaling pathway, the non-canonical arm of NF-κB and other downstream genes causing other morphological changes such as cell proliferation, apoptosis, morphogenesis, and differentiation should be further investigated to elucidate the effect of ORV treatment [37].

Furthermore, ORV also effectively inhibited the production of hBD3 and LL37, antimicrobial peptides secreted by keratinocytes and other innate immune cells when stimulated with bacterial antigens such as LPS or PGN. These antimicrobial peptides have been reported to protect skin from infection by gram-positive and gram-negative bacteria, fungi, and viruses [38,39]. On the other hand, the overexpression of hBD3 and LL37, in turn, stimulates keratinocytes to produce pro-inflammatory cytokines (IL-6, interferon gamma-induced protein (IP)-10) and chemokines (MCP-1, MIP3-α, and RANTES) that play an important role in recruiting activated T cells, macrophages, and other immune cells from the peripheral blood into sites of tissue inflammation, which consequently promote proliferation and inflammation in eczema [39,40,41]. The LL37 and hBD3 production, as well as pro-inflammatory cytokines, may be regulated by the canonical NF-κB p65 signaling pathway. Therefore, ORV treatment can suppress both mechanisms in keratinocytes, according to our findings.

Current treatment approach for eczema includes using of oral or topic agents containing corticosteroids, calcineurin inhibitors, antihistamines, immune suppressants, or biologics and moisturizer [42]. We developed emollient cream containing ORV for topical use and reported that it effectively reduced eczema as evidenced by lower clinical manifestations in DNCB-induced dermatitis mouse model when applied regularly for 10 days. In addition, skin tissues examination also revealed that ORV-containing cream reduced epidermal thickness as well as the number of skin-infiltrating lymphocytes, particularly CD3, CD4, and CD8 T cells. While the CD3 receptor is generally associated with antigen recognition, signal transduction, and activation of immunocompetent T lymphocytes [43], CD4^+^ Th_2_ cells are associated with atopic dermatitis initiation and CD8^+^ T cells play a pivotal role in both contact dermatitis and the chronic phase of atopic dermatitis [44,45]. Our findings that ORV treatment reduced epidermal thickness and skin-infiltrating lymphocytes in vivo clinically prove the anti-proliferation and anti-inflammation properties previously reported. Future studies in humans should be conducted to support the safety and efficacy of ORV therapy on inflammatory skin diseases especially eczematous skin diseases. Finally, we believe we are the first to demonstrate that ORV has an anti-inflammatory effect of ORV that can improve dermatitis and ORV could be utilized as a topical agent for treatment of dermatitis and other inflammatory skin diseases.

## Figures and Tables

**Figure 1 pharmaceutics-15-01709-f001:**
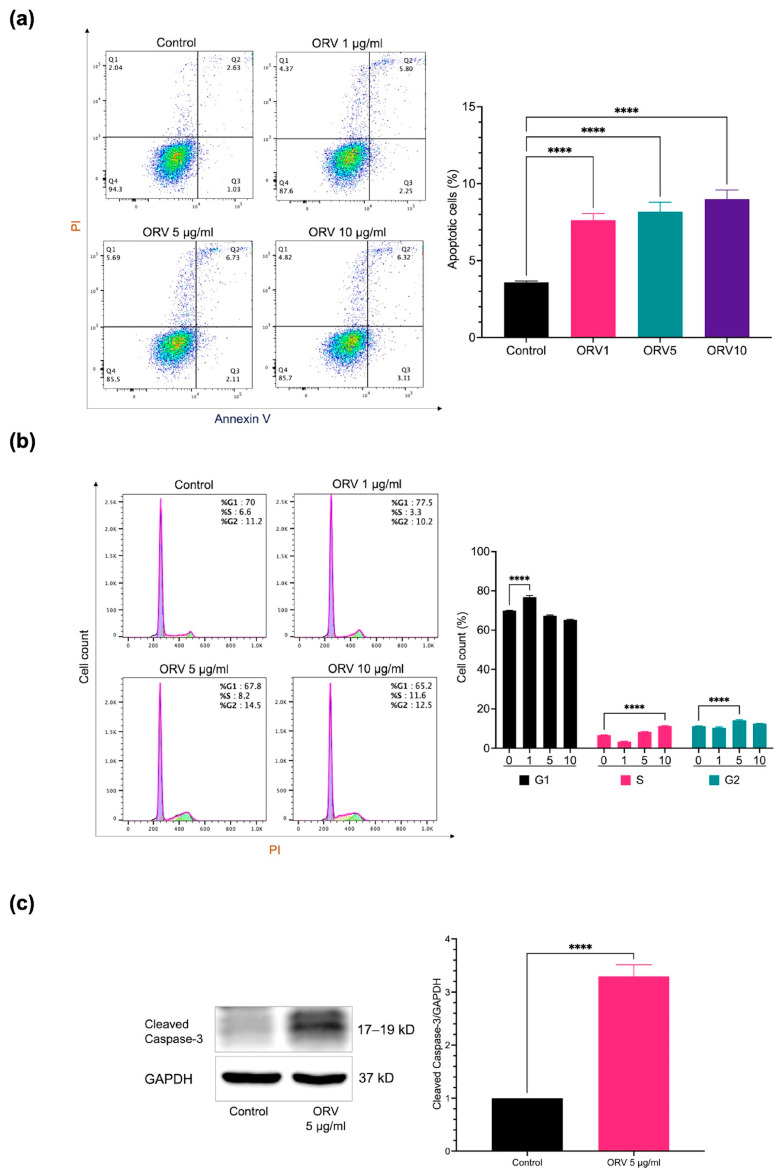
The apoptosis effect of ORV in HaCaT cells. (**a**) The prevalence of apoptotic keratinocytes (Q2 and Q3) after treatment with ORV for 24 h. (**b**) The analysis of cell cycle on HaCaT after treatment with ORV. (**c**) Cleaved caspase-3 proteins (left) and their quantified levels (right). HaCaT cells were treated with ORV 5 μg/mL for 24 h. GAPDH was used as an internal control. Data were present as the mean ± SD. Asterisks denote significant differences compared to untreated cells. **** *p* < 0.0001.

**Figure 2 pharmaceutics-15-01709-f002:**
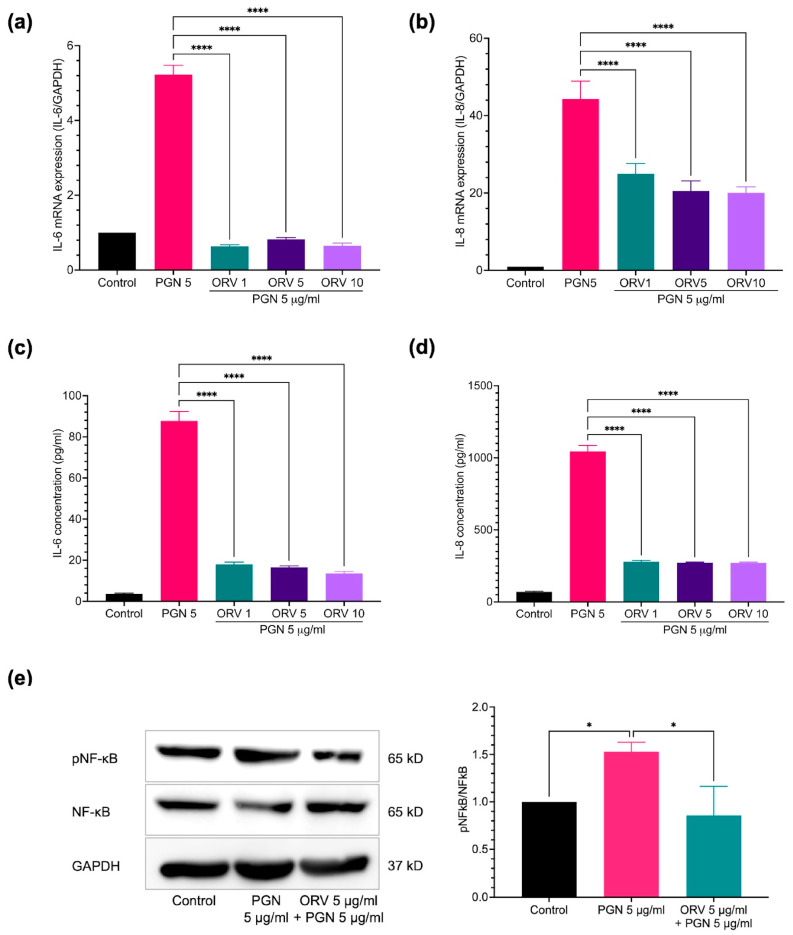
ORV inhibited PGN-induced inflammation in immortalized keratinocytes. HaCaT cells were treated with ORV for 24 h prior to stimulation of 5 μg/mL PGN for another 24 h. Culture supernatants were collected for ELISA analysis and cells were collected for RT-PCR analysis. Expression levels of IL-6 (**a**) and IL-8 (**b**) mRNA and cytokine levels of IL-6 (**c**) and IL-8 (**d**) in the culture supernatants. (**e**) Phosphorylation of NF-κB, total NF-κB, and GAPDH proteins. GAPDH gene and protein are used as internal controls for mRNA and protein expression experiments. Quantitative analysis of protein expression levels is presented as the mean ± SD. * *p <* 0.05; **** *p* < 0.0001.

**Figure 3 pharmaceutics-15-01709-f003:**
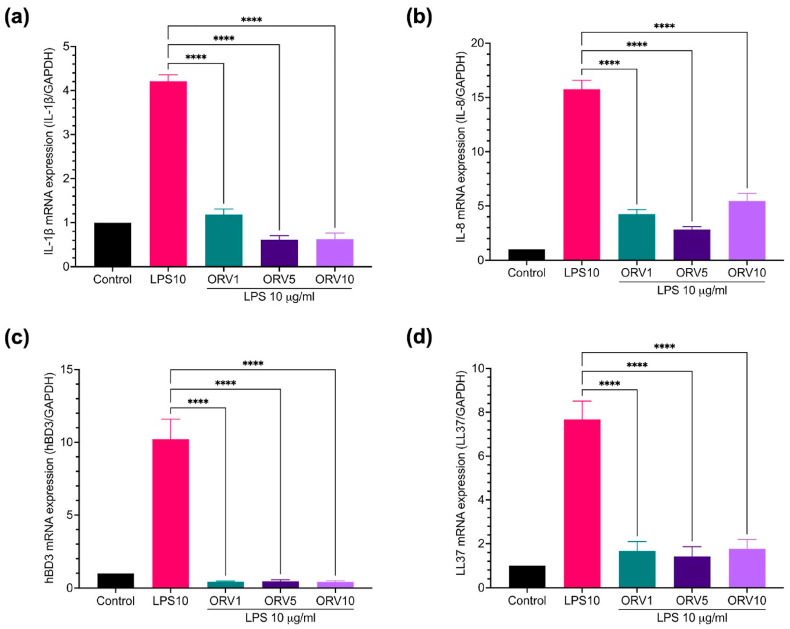
ORV inhibited LPS-induced inflammation in primary human keratinocytes. HEKa cells were treated with ORV for 24 h prior to stimulation of 5 μg/mL LPS for another 24 h. Cells were collected for RT-PCR analysis. Expression levels of *IL-1β* (**a**), *IL-8* (**b**), *hBD3* (**c**), *LL37* (**d**) mRNA are shown. GAPDH was used as a house keeping gene. Asterisks denote significant differences compared to LPS-stimulated cells. **** *p* < 0.0001.

**Figure 4 pharmaceutics-15-01709-f004:**
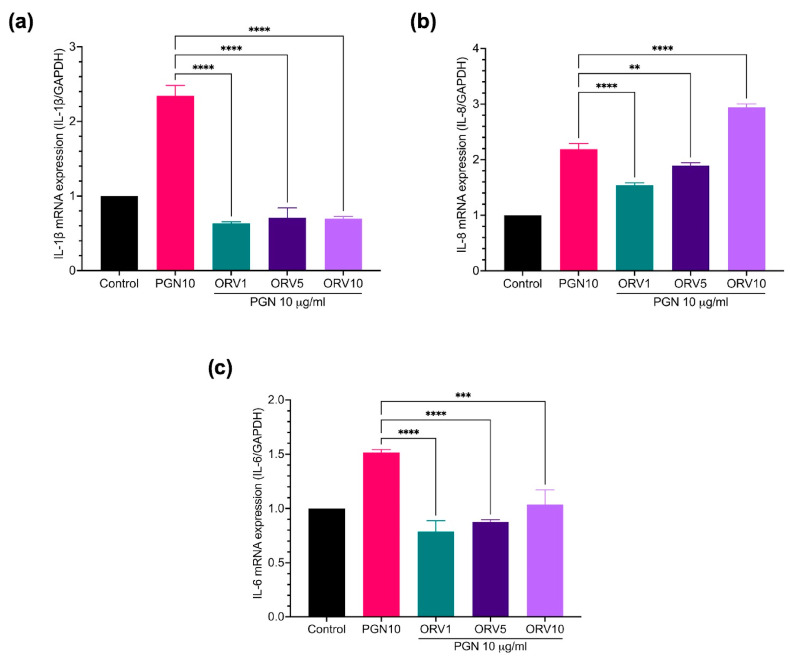
ORV inhibited PGN-induce inflammation in primary human keratinocytes. HEKa cells were treated with ORV for 24 h prior to stimulation of 10 μg/mL PGN for another 24 h. Cells were collected for RT-PCR analysis. Expression levels of *IL-1β* (**a**), *IL-8* (**b**), *IL-6* (**c**) mRNA are shown. GAPDH was used as a house keeping gene. Asterisks denote significant differences compared to PGN-stimulated cells. ** *p* < 0.01; *** *p* < 0.001; **** *p* < 0.0001.

**Figure 5 pharmaceutics-15-01709-f005:**
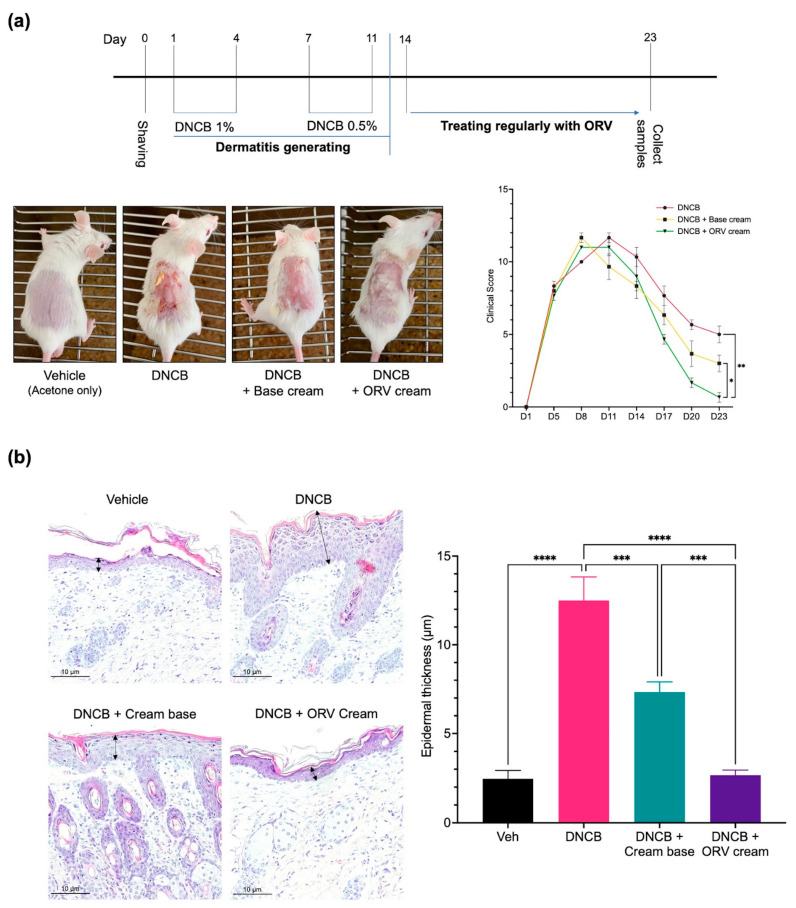
ORV alleviated inflammatory manifestations on mice skin. A schematic diagram of BALB/c mouse skin sensitization using 1% and 0.5% DNCB to induce skin inflammation followed by treatment with ORV cream or vehicle cream (**a**, **top**). Skin images were taken at the end of the experiment (**a**, **bottom left**) and evaluated for clinical skin severity score of dermatitis (**a**, **bottom right**). Histological images of mouse skin epidermis from all treatment groups are shown (200× magnification) (**b**, **left**). Arrows indicated the epidermal thickness. The thickness data from each mouse group in (**b**, **left**) are presented quantitatively as mean ± SD (**b**, **right**). Asterisks denote significant differences compared to DNCB-stimulated group. ***, *p* < 0.001; ****, *p* < 0.0001.

**Figure 6 pharmaceutics-15-01709-f006:**
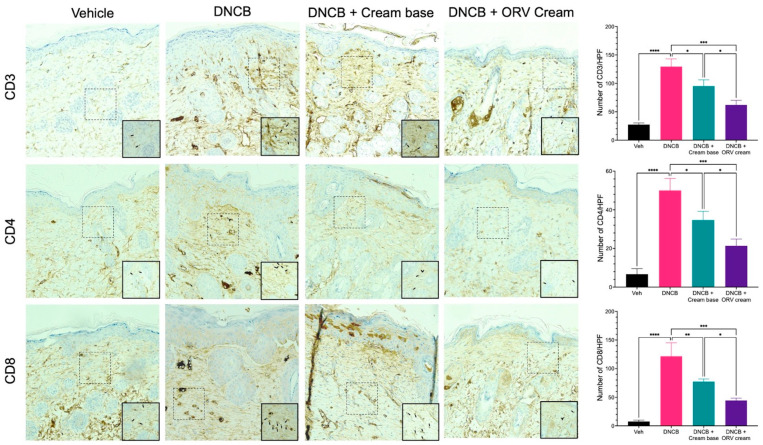
ORV reduced dermal infiltration of inflammatory cells. Cross-sectional images of mouse skins stained with antibodies specific to CD3 (**upper** panel), CD4 (**middle** panel) and CD8 (**lower** panel) to identify CD3^+^, CD4^+^, and CD8^+^ T cells, respectively. The tissue sections were examined under a microscope at a magnification of 400×. The number of cells stained with respective antibody were spotted, counted, and displayed in bar graphs on the right. Quantitative data of T cells are displayed as mean ± SD. Asterisks denote significant differences compared to DNCB-stimulated group. * *p* < 0.05; *** p* < 0.01; *** *p* < 0.001; **** *p* < 0.0001.

## Data Availability

The raw data supporting for the conclusion of this manuscript will be made available by the authors, without undue reservation, to any qualified researcher.

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
