# Peer review of "Oxyresveratrol Attenuates Inflammation in Human Keratinocyte via Regulating NF-kB Signaling and Ameliorates Eczematous Lesion in DNCB-Induced Dermatitis Mice"

_pharmaceutics, 2023, doi:10.3390/pharmaceutics15061709_

Round 1

Reviewer 1 Report

I have carefully reviewed the manuscript titled "Oxyresveratrol Attenuates Inflammation in Human Keratinocyte via Regulating NF-kB Signaling and Ameliorates Eczematous Lesion in DNCB-Induced Dermatitis Mice" and would like to provide my feedback. After thorough consideration, I regret to inform you that I recommend a major revision of the manuscript for publication in its current form. Below, I have outlined the specific reasons for my decision.

My main concern is the inconsistency between in vitro and in vivo models. Authors indicate that inflammation is commonly observed in dermatitis and for this reason the anti-inflammatory capacity of OVR is also investigated with the in vitro keratinocyte models. However, they induce an inflammatory response with the bacterial antigens LPS or PGN whereas the animal model is based in a DNCB-dermatitis model which mimics atopic dermatitis.

From my point of view the manuscript lacks a more in deep discussion on the differences between the inflammatory responses that each model depicts and the implications of these differences on the interpretation of the results. Moreover, recently different in vitro models have been described to mimic atopic dermatitis (Moon S et al., Yonsei Med J. 2021; 62 (11): 969-980; Yang et al., Int J Mol Sci. 2021; 30; 22 (15): 8237. Kang et al. Int J Mol Sci. 2021; 22 (21): 12000) not using bacterial antigens.

A general conclusion that links the in vitro results with the in vivo outcomes would enhance the clarity and scientific rigor of the study.

Other comments

- Why LPS is not also used to stimulate HaCaT cells?

- Lines 233-234: “After determining the optimal concentration for in vitro experiments, we further investigated the proliferation and apoptosis effects of ORV on both types of keratinocytes.” Results obtained in HaCaT cells are presented in Figure 1, but no data is presented for HEKa cells neither in the manuscript or supplementary material.

- Figure 1 caption should be rewritten. Change media for control or untreated cells.

- Figure 2, 3 and 4: Change media for control or untreated cells and change rt-PCR for RT-PCR in figures 3 and 4.

- I suggest the authors to discuss why they have used two keratinocyte cell lines and if one of them is more appropriate or recommended for future research.

P.D. I could not review the supplememntary material, The folder with non-published materials included the Figures of the manuscript.

Author Response

Dear Reviewer,

Reviewer 2 Report

The manuscript describes new properties of oxyresveratrol (ORV), that has been used an antioxidant substance. The authors revealed anti-inflammatory efficacy of ORV on dermatitis models in vitro and in vivo supposing the efficacy is in line with that of glucocorticoid. The authors shown ORV possess apoptosis-induced anti-proliferation property, and inhibited a variety of pro-inflammatory cytokines in keratinocyte in vitro model. They also reveal ORV inhibited the production of antimicrobial peptides secreted by keratinocytes and other innate immune cells when stimulated with bacterial antigens such as LPS or PGN. In 2,4-Dinitrochlorobenzene-induced dermatitis mouse model, ORV treatment reduced lesion severity, and skin thickness and numbers of CD3, CD4 and CD8 T cells in the sensitized skin of mice.

Despite great efforts made by the authors to prove the anti-inflammatory properties of ORV they surprisingly use in the experiments not the pure substance 2,4,3’,5’-tetrahydroxystilbene but a mixture of likely biologically active components extracted from Artocapus lakoocha, containing on around 10% additional substances. The resulting action of such biologically active additive was analyzed in the manuscript, however nobody knows the primary action substance. The authors have to carry out the experiments using chemically pure ORV substance, or present detailed chemical analysis of the mixture used in the experiments. Otherwise a sentence described the situation must be added in the manuscript. Also, the conclusion made by authors at lines 428-429 should be reformulated less ambitiously.

Since the authors prove the mixture contain ORV considerably inhibited the production of LL37 simultaneously elevated the level of activated caspase-3, one of the ORV mixture cream application may be rosacea treatment. The inhibition of mTORC1 pathway maybe additionally discussed in the manuscript in this the angle of view.

Line 407: please reformulate bearing in mind that IL-10 is rather anti-inflammatory, not pro-inflammatory cytokine, though may reveal both activities.

Personal pronouns should be removed from the manuscript: eg. In the abstract section instead of ‘we examined’, write " ‘was examined’, instead of ‘we conducted’, write ‘was conducted’, instead of ‘we demonstrated’ write ‘it has been demonstrated’.

Author Response

Dear Reviewer, 

Reviewer 3 Report

1. In the abstract, introduce what the abbreviation DNCB stands for

2. Line 25, and line 43 have the same abbreviation one letter is missing in the abbreviation, and line 46 is the same...watch out for such technical errors

3. Why did you write in the first person plural?

4. Think about reformulating the title somehow, it has no strength

5. Line 38 the cell, line 44 the LDS blasphemy.... check the members throughout the work

6. Lines 81 or 82, or you will state, as with some, the specific name of the microorganism or just the genus, but then I think it should go sp. Check it out.

7. In the end, emphasize how the ORV was obtained, whether you use the whole extract for testing or whether you isolated it, be a little more precise

8. The introduction and the objective are very well written.

9. Line 99 uses the SI system of units for hours, eg. (in full paper)

10. Line 96, 103 ° is in superscript (throughout the work)

11. 2.2. why DMSO when it has specific effects on the skin itself? Also, is there a possibility to make references here whether work, technical solution, or patent. ... to have a closer picture of what is being done as soon as possible?

12. 2.4 Couldn't this test be performed on skin preparation and skin cell culture tests, eg 32°C?

13. In the materials, please highlight and reference everything taken from somewhere, modified or otherwise.

14. The results, discussion, and conclusion are excellently presented.

1. In the abstract, introduce what the abbreviation DNCB stands for

2. Line 25, and line 43 have the same abbreviation one letter is missing in the abbreviation, and line 46 is the same...watch out for such technical errors

3. Why did you write in the first person plural?

4. Think about reformulating the title somehow, it has no strength

5. Line 38 the cell, line 44 the LDS blasphemy.... check the members throughout the work

6. Lines 81 or 82, or you will state, as with some, the specific name of the microorganism or just the genus, but then I think it should go sp. Check it out.

7. In the end, emphasize how the ORV was obtained, whether you use the whole extract for testing or whether you isolated it, be a little more precise

8. The introduction and the objective are very well written.

9. Line 99 uses the SI system of units for hours, eg. (in full paper)

10. Line 96, 103 ° is in superscript (throughout the work)

11. 2.2. why DMSO when it has specific effects on the skin itself? Also, is there a possibility to make references here whether work, technical solution, or patent. ... to have a closer picture of what is being done as soon as possible?

12. 2.4 Couldn't this test be performed on skin preparation and skin cell culture tests, eg 32°C?

13. In the materials, please highlight and reference everything taken from somewhere, modified or otherwise.

14. The results, discussion, and conclusion are excellently presented.

Author Response

Dear Reviewer, 

Reviewer 4 Report

The manuscript entitled " Oxyresveratrol Attenuates Inflammation in Human Keratinocyte via Regulating NF-kB Signaling and Ameliorates Eczematous Lesion in DNCB-Induced Dermatitis Mice" presented by Tran et al, summaries a research pertaining to effect of Oxyresveratrol on dermatitis via NF-kB regulations and other associated mechanism. Overall manuscript sounds good and delivering the scientific contents. However, some major exercise is required to improve further its content:   

·       In abstract please provide the idea to select Oxyresveratrol for this study. May be little background.

·       All abbreviation should be spelled on first time citation like DNCB.

·       Is Artocarpus lakoocha is only sources of Oxyresveratrol, please provide the details if any others.

·   It would be better if authors provide chemical structure of Oxyresveratrol or chemistry with reference to resveratrol.

·       In vitro must be italic (Line 89, Page 2). DO similar correction throughout the MS

·       Please add material sections separately, where all details related to procurement of material such as cell lines, chemical and other must incorporated.

·       In discussion “ORV has long been reported as an antioxidant reagent with potential dermatological applications”. Please support these lines with suitable references.

·       Discussion can be improved and the findings must be discussed with similar findings in details.

·       Any other future studies that need to warranted must be concluded for conversion of these findings to some value added products.

·       Language and any other typological mistake can be address

·       Please check pattern of reference as per format.

Language and any other typological mistake can be address

Author Response

Dear Reviewer, 

Round 2

Reviewer 1 Report

Dear authors,

thank you for the answers. I only has a final comment related to lines 412-413: Because the immortalized cell line is the cell undergoes in vitro modification. It may have some different properties from primary keratinocyte cell. Please rewrite the sentences for example as: The immortalized cell line, resulting from in vitro modification, may exhibit distinct properties compared to primary keratinocyte cells.

Yours sincerely,

Reviewer 2 Report

Thank you for the corrections made.

Reviewer 4 Report

Manuscript is improved now and authors resolved all my queries.